# Improving lifestyle behaviours among women in Kisantu, the Democratic Republic of the Congo: A protocol of a cluster randomised controlled trial

Diana Sagastume[1]*, Deogratias Katsuva Sibongwere[1], Olivier Kidima[2,3], Diertho Mputu Kembo[4], José Mavuna N'keto[5], Jean-Claude Dimbelolo[6], Dorothée Bulemfu Nkakirande[7], Jean Clovis Kalobu Kabundi[2,3], José L. Peñalvo[1]

1 Department of Public Health, Non-Communicable Diseases Unit, Institute of Tropical Medicine, Antwerp, Belgium, 2 Memisa, Brussels, Belgium, 3 Memisa Representation in Kinshasa, Kinshasa, Democratic Republic of the Congo, 4 BDOM-Kisantu Centre Pastoral/Mission Catholique Kisantu, Kisantu, Democratic Republic of the Congo, 5 Health District of Kisantu, Kisantu, Democratic Republic of the Congo, 6 Centre d'éducation Diabète & Santé, Kinshasa, Democratic Republic of the Congo, 7 Division des Maladies Non Transmissibles, Direction Surveillance Epidémiologique, Ministère de la Santé, Kinshasa, Democratic Republic of the Congo

* dsagastume@itg.be

**Data Availability Statement:** Currently, datasets have not been yet generated or analysed as this manuscript describes the study protocol of

## Abstract

### Introduction

As the prevalence of obesity among women of reproductive age is increasing in sub-Saharan Africa, the burden of lifestyle-related conditions is expected to rise quickly. This study aims to develop and evaluate a multi-component health promotion programme for a healthy lifestyle to ultimately prevent the onset of type 2 diabetes and gestational diabetes among adult women in Kisantu, the Democratic Republic of the Congo.

### Methods and analysis

This study is a cluster randomised controlled trial whereby two groups of three healthcare centres each, matched by population size coverage and geographical area, will be randomised to an intervention or a comparison group. Adult women of reproductive age (18–49 years), non-pregnant or first-trimester pregnant, will be recruited from the healthcare centres. 144 women in the intervention centres will follow a 24-month multi-component health promotion programme based on educational and motivational strategies whereas the comparison centres (144 participants) will be limited to a basic educational strategy. The programme will be delivered by trained peer educators and entails individualised education sessions, education and physical activity group activities, and focus groups. Topics of an optimal diet, physical activity, weight management and awareness of type 2 and gestational diabetes will be covered. The primary outcome is the adherence to a healthy lifestyle measured by a validated closed-ended questionnaire and secondary outcomes include anthropometric measurements, clinical parameters, diet diversity and the level of physical activity. Participants from both groups will be assessed at baseline and every 6 months by trained

ongoing research. After the study termination, data cannot be shared publicly because it entails sensitive information, however, data generated will be available upon request to the corresponding author and only after consultation with the involved ethical committees. Both the Institutional Review Board of the Institute of Tropical Medicine (contact via: irb@itg.be) and the Ethical Committee of the University of Kinshasa (contact via: espsec_unikin@yahoo.fr) should review the request and approve any use of the data generated from this project outside the original scope of work, and identified research teams.

**Funding:** This work was supported by was the City of Antwerp, Belgium; reference number: 436261/30/10. This funding source had no role in the design of the study and will not have any role during the execution, analyses, interpretation of the data or decision to submit results.

**Competing interests:** All authors declare that they have no competing interests.

health professionals from the recruiting healthcare centres. Data will be summarised by measures of central tendency for continuous outcomes, and frequency distribution and percentages for categorical data. The primary and secondary outcomes will be quantified using statistical mixed models.

## Ethics

This research was approved by the Institutional Review Board of the Institute of Tropical Medicine Antwerp in Belgium (IRB/RR/AC/137) and the Ethical Committee of the University of Kinshasa in the Democratic Republic of the Congo (ESP/CE/130/2021). Any substantial change to the study protocol must be approved by all the bodies that have approved the initial protocol, before being implemented. Also, this journal will be informed regarding any protocol modification. Written informed consent will be required and obtained for all participants. No participant may be enrolled on the study until written informed consent has been obtained.

## Trial registration number

NCT05039307.

## Introduction

According to the World Health Organisation (WHO), the global prevalence of obesity has nearly tripled over the last decades, with 650 million adults considered obese in 2016, among more than 1.9 billion overweight adults, and with substantially larger increases in low- and middle-income countries (LMICs) [1]. In Africa, there is an overall growing upward trend in body weight and adiposity [2], driven by epidemiological and nutritional transitions towards unhealthy lifestyles, rapid urbanisation and population ageing [3]. Particularly in most sub-Saharan Africa (SSA) nations, women have been reported to have a higher risk of overweight and obesity than men [4, 5] and, concerningly, obese (body mass index (BMI) $> = 30$ kg/m$^2$) women have a 28-times higher risk of developing type 2 diabetes (T2D) than those with a normal weight [6]. This increased risk of obesity among females in SSA is of particular importance among women of reproductive age [7], as excess weight before or during pregnancy predisposes women to develop gestational diabetes mellitus (GDM), a transitory but serious condition associated with negative health outcomes for both the mother and the offspring [8]. The prevalence of maternal (pre-pregnancy or first trimester) obesity in SSA is reported to vary widely between 6.5% and 50.7%, being more frequent among older and multi-parous women [9] and, in Central African countries, a prevalence of GDM of 20.4% has been estimated [10]. Thus, preventive strategies for both conditions, T2D and GDM, warrant urgent consideration.

With overlapping risk factors, T2D and GDM can both be prevented or delayed by adhering to an optimal lifestyle [11–13]. Studies have suggested that lifestyle modification strategies, including those instilling adequate dietary and physical activity habits accounting for culturally relevant conditions, are effective for preventing lifestyle-related conditions mainly through weight control [14, 15]. Similarly, there is solid evidence of the positive effects of maintaining an optimal diet and exercise during pregnancy in the reduction of the risk of GDM [11]. Also, it has been highlighted the need for comprehensive (diet and physical activity) preventive strategies addressing primary prevention of T2D and GDM, among women of reproductive age as they, and their offspring, are particularly vulnerable groups for these conditions [10,

16]. Therefore, to ultimately prevent the onset of lifestyle-related conditions, particularly T2D and GDM, this study aims to develop and evaluate the impact of a multi-component health promotion programme to encourage a healthy lifestyle among adult women of reproductive age in Kisantu, the Democratic Republic of the Congo (DRC).

## Methods/Design

### Objectives

To reduce the incidence of T2D and GDM in a long term, the overall aim of this study is to develop, implement and evaluate a multi-component health promotion programme for healthy living among adult women of reproductive age in Kisantu, DRC. To achieve this over-all objective we aim to increase the number of women with a healthy lifestyle, reduce the incidence and prevalence of overweight and obesity among this population and increase the number of women with a healthy weight gain during pregnancy corresponding to their pre-pregnancy BMI. Other supportive objectives embedded in this research include the improvement of the usability of data capturing systems for participant's follow-up and strategy-adaptation based on data analysis; improving the knowledge among health professionals on effective strategies for the prevention of T2D and GDM and their risk factors; improve the technique and precision of measurements concerning health visits among health professionals; improve women's attendance to antenatal care (ANC) and therefore have an impact on maternal and neonatal health; promote a healthy lifestyle through the support of community activities.

### Evaluation design

This study is a 2-arm parallel cluster randomised controlled trial (cRCT) whereby two groups of three healthcare centres were randomised to an intervention or a comparison group. The intervention group will be provided with a 24-month multi-component health promotion programme based on educational and motivational strategies whereas the comparison group will be limited to a basic educational strategy only. We have set the duration of the programme to 24 months to guarantee we have sufficient time to witness an improvement in lifestyle and weight control in the participants of the intervened clusters. This trial began in October 2021 and will be finalised in October 2023; recruitment is complete and the study is ongoing. Fig 1 shows the SPIRT diagram for the allocation, enrolment, intervention and assessments according to the most recently approved version of the protocol (current V 2.1; June 2021).

### Randomisation and blinding

Pair matching randomisation will be used to ensure the two groups of three healthcare centres were balanced in terms of population size coverage and geographical area (rural/urban). Such randomisation will be performed manually by creating the two groups of three healthcare centres with matching characteristics followed by randomly assigning the multi-component health promotion programme to one group using a random allocation generator (Fig 2). Investigators of the study will carry out the randomisation process.

Due to the nature of this research and the impossibility of masking the programme, this study will not be blinded.

### Setting

Kisantu district is a semi-urban zone located in the province of Kongo Central in the southwest of DRC with, last estimated in 2015, nearly 150,000 inhabitants [17], however, it is expected to reach ~202,000 inhabitants currently. This study will be carried out in six

| | STUDY PERIOD | | | | | | |
|---|---|---|---|---|---|---|---|
| | Allocation | Enrolment | Post allocation and enrolment | | | | Close-out |
| Timepoint | -T2 | -T1 | T0<br>Baseline | T1<br>(6 months) | T2<br>(12 months) | T3<br>(18 months) | T4<br>(24 months) |
| **Enrolment** | | | | | | | |
| Eligibility criteria | | X | | | | | |
| Informed consent | | X | | | | | |
| Allocation of intervention | X | | | | | | |
| **Intervention** | | | | | | | |
| Multi-component programme | | | X | ← | | → | X |
| Comparison intervention | | | X | | | | |
| **Assessments** | | | | | | | |
| **Demographic characteristic** | | | | | | | |
| Age | | | X | | | | |
| Marital status | | | X | | | | |
| Profession | | | X | | | | |
| Profession of spouse | | | X | | | | |
| Number of children | | | X | | | | |
| Number of people living in the household | | | X | | | | |
| Smoking status | | | X | | | | |
| Smoking status of spouse | | | X | | | | |
| Alcohol consumption | | | X | | | | |
| Pre-pregnancy weight | | | X | | | | |
| Allergies | | | X | | | | |
| **Primary outcome** | | | | | | | |
| Adherence to a healthy lifestyle | | | X | X | X | X | X |
| **Secondary outcomes** | | | | | | | |
| Height | | | X | | | | |
| Weight | | | X | X | X | X | X |
| Body mass index | | | X | X | X | X | X |
| Waist circumference | | | X | X | X | X | X |
| Glycemia in blood | | | X | X | X | X | X |
| Systolic and diastolic blood pressure | | | X | X | X | X | X |
| Diet diversity | | | X | X | X | X | X |
| Level of physical activity | | | X | X | X | X | X |
| **Others** | | | | | | | |
| Acceptability of community activities | | | X | X | | | |

**Fig 1. SPIRIT flow diagram of enrolment, intervention and assessment.**

healthcare centres from Kisantu Health District covering approximately 107,000 inhabitants, sharing an integrated strategy ANC and providing care to patients with T2D.

## Population

Participants will be recruited primarily in the healthcare centres by the health professionals, as women attend frequently due to general consultations for ANC, post-natal care, child

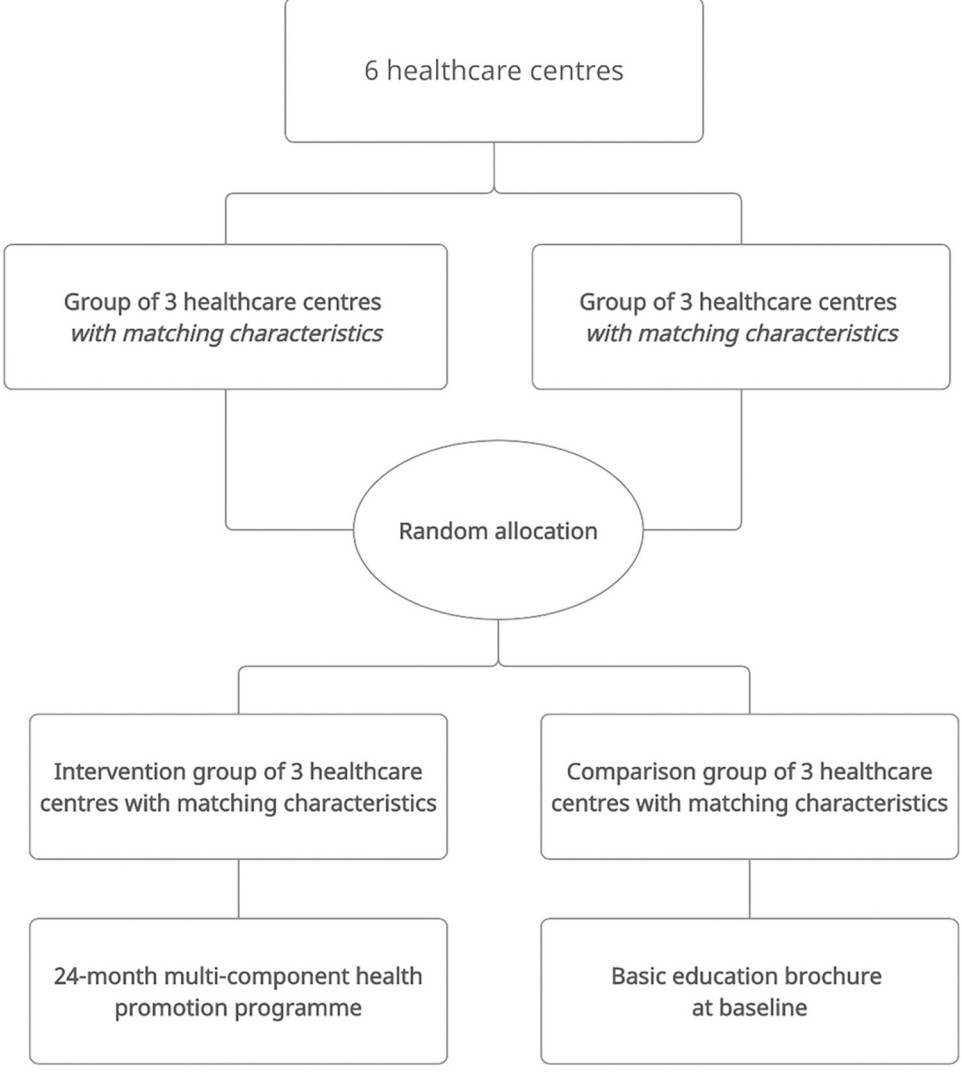

Matching characteristics indicate both groups are balanced in terms of population size coverage and geographical area (rural/urban).

**Fig 2. Study design and randomization.**

monitoring, and vaccinations, among others. In the recruitment process, health professionals will identify potential participants and provide them with information about the study. If interested, participants will be assessed for compliance with the inclusion criteria and requested to provide written informed consent (IC). To be eligible participants must be adult women of reproductive age (18–49 years old) non-pregnant or first-trimester pregnant without a previous diagnosis of type 1 diabetes or T2D (confirmed by a random glucose test carried out during the recruitment process). The exclusion criteria include males, children and adolescents of both sexes (<18 years old), and pregnant women with any type of diabetes; participants with pre-existing severe chronic conditions, such as chronic kidney or liver disease; living outside of the surrounding catchment areas of the 6 healthcare centres; women internally displaced, without a permanent home or identified physical address; women planning to move elsewhere in the following 2 years; and women that do not provide a written IC.

## Multi-component health promotion programme

The programme will be built on health promotion strategies for T2D and GDM as advised by the WHO [18]. A multi-component health promotion programme will be developed based on educational and motivational strategies and will be delivered by trained peer-educators from the women's community. Because of the challenges associated with behaviour change, motivational strategies have been identified as an effective means to enhance interventions that aim to promote healthy dietary and/or physical activity habits [19]. During the first year of the programme, the possibility of community activities engagement will be explored, as community participation is a fundamental element for positive public health impact, as well as in reducing the stigma against T2D [20, 21]. If acceptable, community activities will be implemented in the second year of the programme. Programmes based on a multi-component approach have been proven to be the most effective preventive strategies [22], particularly when targeting diet and/ or physical activity [23].

The topics and components of the programme will be selected by the health professionals and nutritionists in the research team, after discussing the relevance to the population's context and agreed upon by all investigators. International recommendations will be the basis of the programme, mainly from the WHO [1, 18, 24–26], the American Heart Association [27–30] and the Harvard T.H. Chan School of Public Health [31], among other international public health institutions [32–35], and will be slightly adapted to the local context, particularly the food items, by health professionals of the Kisantu health area. The topics will include A) Healthy diet: food pyramid, healthy plate portions distribution, type of carbohydrates and fats, healthy cooking techniques including healthy oils; B) Physical activity: definition, types of physical activity, advantages and benefits, examples, and practical physical activity sessions; C) Weight control: strategies, benefits and healthy weight gain during pregnancy; and D) Awareness of T2D and GDM: definition, prevention, management and complications (Table 1). The multi-component programme will consist of 1) four individualised education visits of 30 minutes provided at the participant's household; and 2) four group activities of 60 minutes duration (one group education session, one motivational group session and two physical activity sessions) organised in the proximity of the healthcare centres. The motivational components of goal setting and the use of the pedometer are incorporated in the individualised education visits. These components will be distributed and carried out during the 24-month programme (Fig 3). Participants will be provided with printed materials, and a reference healthy lifestyle brochure, both at baseline and during the individual education sessions at the household containing a summary of the topics discussed. Additionally, as a process evaluation of the programme, four 60-minute focus groups will be carried out at the healthcare centres aiming at assessing the perception of the programme and identifying barriers to be further addressed.

The comparison group will be limited to receiving only the reference healthy lifestyle brochure with general recommendations during the baseline assessment.

## Outcomes

To directly evaluate the impact of the health promotion programme aiming at instilling healthy living, to ultimately dimmish modifiable risk factors of T2D and GDM, the primary outcome of this study is the adherence to a healthy lifestyle measured by a validated closed-ended questionnaire adapted to the context. This primary outcome was also chosen due to the study's particular focus on health promotion, particularly by approaching public health from an upstream perspective and expecting to counteract the initiation of unhealthy lifestyle habits, such as sub-optimal diet and physical inactivity. The secondary outcomes include the anthropometric measurements of weight, height, calculated BMI, and waist circumference as well as

**Table 1. Description of the multi-component health promotion programme.**

| Component | Topics |
|---|---|
| *Individualised education (E) and motivation (M)* | Visit 1:<br>• (E) Optimal diet: the food pyramid adapted to DRC<br>• (E) Optimal diet: plate distribution<br>• (M) Setting goals and use of pedometer explanation<br>Visit 2:<br>• (E) Physical activity: definition, recommendations and advantages of physical activity<br>• (E) Weight control: definition, causes, consequences and prevention of obesity.<br>• (M) Discuss the progress of the goals and use of the pedometer<br>Visit 3:<br>• (E) Optimal diet: type of carbohydrates and fats<br>• (E) Optimal diet: healthy cooking techniques<br>• (M) Discuss the progress of the goals and use of the pedometer<br>Visit 4:<br>• (M) discuss the progress of the goals and use of the pedometer |
| *Group education (E), Motivation (M) and activities (A)* | Session 1<br>• (E) Optimal diet: food security<br>• (E) Weight control: healthy weight gain during pregnancy<br>• (E) Physical activity: Explanation of the types of physical activity<br>• (A, M) 2 series of 15 minutes of walking and 5 minutes of muscle training.<br>Session 2<br>• (E) Definition, causes, risk factors, complications and prevention of T2D and GDM<br>Session 3<br>• (A, M) Sharing experiences for the identification of obstacles and opportunities for health promotion<br>Session 4<br>• (A, M) 2 series of 15 minutes of walking and 5 minutes of muscle training. |

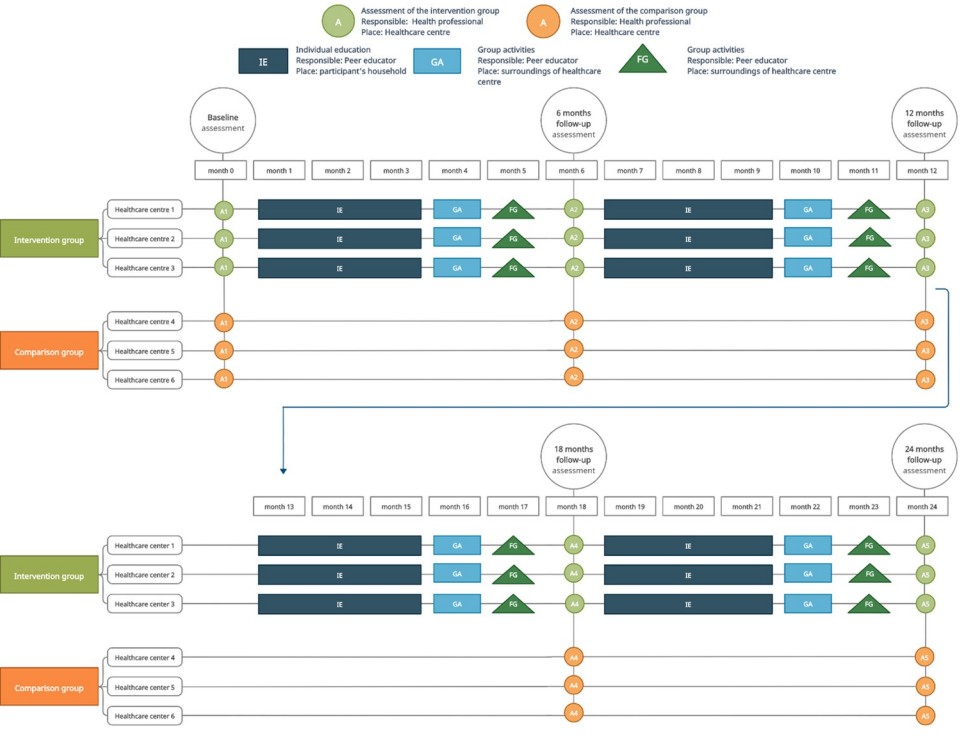

**Fig 3. Timeline of the multi-component health promotion programme.**

the clinical indicators of systolic blood pressure (SBP), diastolic blood pressure (DBP) and fasting glycemia in blood. In addition, diet diversity and the level of physical activity will be evaluated. All outcomes will be assessed in all participants from both groups except for waist circumference in pregnant women.

## Variable collection

At baseline, demographic variables will be collected including age, tobacco and alcohol use, pre-pregnancy weight (exclusively for pregnant women), and known allergies, as well as profession, marital status, spouse's profession (if married/partnered), number of children, and number of people living in the household. Both the primary and secondary outcomes will be measured at baseline and thereafter every 6 months, providing 5-time points per participant in ideal circumstances. The baseline and follow-up assessments will be conducted by the trained health professional at each healthcare centre.

The adherence to a healthy lifestyle will be measured by a closed-ended multiple-choice questionnaire including questions regarding diet and physical activity. The questionnaire was developed in India by Dubasi and co-workers to assess the adherence to dietary and physical activity advice for weight control in lifestyle-related diseases [36]. This tool comprises 14 questions, 12 regarding diet and 2 related to physical activity. Each question has 5 multiple choice answers entailing different frequencies of lifestyle behaviours. The answers to each question will be weighted and assigned with a score based on the relevance and interest of this study, providing a minimum response of 1 and a maximum of 5 points per question. The final score will range between 14–65 points, with lower scores indicating poor adherence to a healthy lifestyle, while a higher score suggests good adherence. The questionnaire has been adapted to the context of DRC by the local health professionals and research team (S2 File).

The anthropometric measurements will be performed according to the guidelines for the collection of physical measurements advised by WHO's STEPwise approach to non-communicable disease (NCDs) risk factor surveillance [37]. Height will be measured in centimetres (cm) using a height measuring board (ADE® Germany GmbH) positioned on a firm surface against a wall. Participants will be asked to remove shoes and headgear, stand on the board facing the back towards the measuring board with feet together, heels against the backboard, knees straight and look straight ahead. The measuring arm will be moved gently until it compresses the hair of the participant, subsequently, the measure will be recorded. Weight will be recorded in kilogrammes (kg) and assessed by a mechanical weight scale (ADE® Germany GmbH) placed on a firm and flat surface. Participants will be asked to remove any footwear and socks, empty pockets and remove any heavy objects like belts, mobiles, wallets, etc. They will be asked to step onto the scale, stand still, face forward, place arms on the side, and wait until asked to step off after the measure has been recorded. BMI will be calculated by dividing the weight in kilogrammes by the height in metres squared. Waist circumference will be measured in centimetres with a measuring tension tape. Ideally, it will be measured without clothing, however, if not possible, the measurement may be taken over light clothing. This measurement will be assessed in cm and the reader will be done at the level of the tape to the nearest 0.1 cm, making sure the tape is snug but not tight enough to compress the skin.

Clinical parameters will be assessed following also the guidelines for the collection of physical measurements advised by the WHO STEPwise approach [37]. Glycemia will be assessed by random glucose tests. The device, glucometer (Medisign® GmbH, Germany), instructions and hygienic recommendations will be followed. DBP and SBP evaluation will be via a digital automatic blood pressure monitor (Sanitas® Germany) and following the instructions of the device and cuff application. First, the health professional will place the left arm of the

participant on a table with the palm facing upwards, followed by removing or rolling up the clothing on the arm. Later, the appropriate cuff size will be chosen and positioned 1–2 cm above the elbow joint, then the cuff will be wrapped and securely fastened with Velcro. Second, following the instructions of the device, the monitor will be switched on and will start measuring the blood pressure when the pulse has been detected. The SBP and DBP should be displayed in a few seconds, thereafter the readings will be recorded in mmHg.

Diet diversity will be assessed by using the minimum dietary diversity for women (MDD-W) developed by the Food and Agriculture Organisation (FAO) of the United Nations [38]. The MDD-W is used as a proxy to describe one important dimension of women's diet quality based on micronutrient adequacy. This questionnaire is a dichotomous indicator of whether participants have consumed at least five out of ten defined food groups the previous day. The ten defined groups include 1) Grains, white roots and tubers, and plantains; 2) Pulses (beans, peas, and lentils); 3) Nuts and seeds; 4) Dairy; 5) Meat, poultry, and fish; 6) Eggs; 7) Dark green leafy vegetables; 8) Other vitamin A-rich fruits and vegetables; 9) Other vegetables; 10) Other fruits.

The level of physical activity of the participants will be assessed by using the short version of the International Physical Activity Questionnaire (IPAQ) [39]. This questionnaire reflects the level of physical activity through the evaluation of the frequency, duration and type of physical activity performed. This tool comprises seven questions and collects information on physical activity involved in the domains of leisure time, domestic and gardening activities, and work-related and transport-related activities. Consequently, the Metabolic Equivalent (MET) per week will be calculated allowing to determine and classify the weekly level of physical activity (low/medium/high) of the participant.

To explore the feasibility of complementing the multi-component intervention programme with community-based activities, closed-ended Likert scale questionnaires with an agreement scale (Strongly disagree (1), disagree (2), undecided (3), agree (4), strongly agree (5)) will be performed twice during the first year of intervention. If acceptable and feasible, activities engaging the community will be integrated into the intervention programme and delivered exclusively to the intervention group. These activities will be guided by a designated volunteer among the community leaders, and preferably a participant in the intervention group. The designated community leader will oversee organising and promoting healthy lifestyle activities with support from the peer educators and other involved team members. Potential activities to be integrated include, but are not limited to, promoting physical activity (e.g. walking pathway), sharing healthy cooking recipes and experiences, and healthy lifestyle education involving children and family, among others.

## Quality assurance

This study will be monitored following regulations applicable to clinical trials and standard operation procedures (SOPs) for training, recruitment, intervention programme, assessments and data management developed by the investigators. The quality of the study implementation will be supervised by the principal investigators every 3 months and monthly by the local project managers. Additionally, this study has the support of the Health Ministry of DRC and personnel from Kisantu's health district are involved in the monitoring of the study.

Procedures for potential drop-outs are envisioned. When the health professional or peer educator has no news of the participants after six months (follow-up assessment), the health professional must make every effort to contact the participant by telephone to determine the reason for missing the follow-up assessment. If no news of the participant for one year and/or the participant missed two follow-up assessments, the health professional must make every

effort again to contact the participant to establish the reason for the discontinuation of the study and to suggest and request the participant to attend an end-of-study visit that will include the final assessment. If all these attempts to contact the participant fail, the investigator can then declare the participant "drop-out".

To ensure high-quality research, different steps have been undertaken. Project managers were firstly trained by principal investigators in all aspects of the study including adherence to the protocol and procedures, assessment of outcomes and intervention programme procedures and delivery. Furthermore, the project managers will recruit 12 health professionals, 2 from each healthcare centre, and provide a 5-day training on their roles and responsibilities, adherence to the protocol, and assessment of outcomes. Similarly, 6 peer educators, 2 from each healthcare centre from the intervention group, will be trained also in their roles and responsibilities, adherence to the protocol and particularly on the intervention programme procedures and delivery. Both, health professionals and peer educators, will be provided with guidelines including all the procedures of the study. If needed, additional training for reinforcing adherence to the protocol, assessments or intervention procedures and delivery will be provided throughout the study period.

Data quality control will be performed every 6 months after each assessment has been carried out. Data will be assessed for completeness, consistency, accuracy, and cleanness. Moreover, the information collected by the focus groups will provide the investigators and project managers with a qualitative assessment of the intervention from the point of view of the participants which will be discussed and considered for further improvements.

## Power calculation

Evidence measuring changes in lifestyle habits using a similar score scale has reported a 20% differential improvement between intervention and comparison groups from baseline to follow-up after 8 months [40]. Moreover, interventions concerning dietary and physical activity improvements have described intra-class correlation coefficients (ICC) ranging between 0.007–0.3 [41, 42]. In line with the previous evidence and the demographics of the study, we have conducted a power estimation based on the assumptions of 144 participants per arm (288 participants in total divided into 6 healthcare centres), a target improvement of 12% (equivalent to + 6 score points) in the primary outcome between the groups, effect size of 0.7, ICC of 0.1, and 20% loss to follow-up to consider individual drop-outs. Based on the previously stated and considering a 95% level of significance (one-sided), the estimated sample size would provide a power of 91% to detect the improvement foreseen. Considering the same scenario plus the possibility of the withdrawal of one healthcare centre, a power of 85% would be reached. The power calculation was performed in STATA with the specification for cluster-randomised designs.

## Data analysis plan

After the initiation of the study, demographic characteristics will be reported as measures of central tendency (mean and SD or median and interquartile range) for continuous outcomes and frequency distribution and percentages for categorical data. Group differences in baseline values and demographics will be compared with t-tests or chi-square statistics.

The analyses will be conducted based on the intention-to-treat principle. Primary and secondary outcomes will be evaluated for normality by the Kolmogorov-Simonov test and visual inspection of histograms. To evaluate whether the intervention group has a greater improvement in the lifestyle questionnaire score than the comparison group after the intervention programme, we will use mixed models for longitudinal data to account for repeated

measurements of the primary outcome, where the intervention group will be specified as a fixed effect and adjusted for the relevant covariates, and healthcare centres will be handled as random effects; unstructured covariance matrix will be used. To assess the changes in the secondary outcomes, the same approach will be used for anthropometric measurements, clinical parameters, diet diversity and level of physical activity. Potential exploratory analyses include subgroup analyses based on pregnancy status, and categories of lifestyle adherence score, BMI and glycemia.

A $p < 0.05$ will be considered statistically significant for all tests. Multiple comparisons will be adjusted by Bonferroni's method. Missing data will be handled by multiple imputations, if necessary. Data will be captured in a Microsoft Excel database template developed by the Institute of Tropical Medicine. All analyses will be conducted using STATA (Release 16/SE. College Station, TX: StataCorp LP).

## Ethics statement

This research was approved by the Institutional Review Board of the Institute of Tropical Medicine Antwerp in Belgium (IRB/RR/AC/137) and the Ethical Committee of the University of Kinshasa in DRC (ESP/CE/130/2021). Any substantial change to the study protocol must be approved by all the bodies that have approved the initial protocol, before being implemented. Also, this journal will be informed regarding any protocol modification. Written informed consent will be required and obtained for all participants. No participant may be enrolled on the study until written informed consent has been obtained.

## Confidentiality and risks

All information will be kept confidential and securely stored. The privacy and confidentiality of the participants will be assured by anonymous IDs and coding techniques for traceable information. All study-related documents and data will be stored in a secured location and access will be granted to only those involved in the research.

Participants are subject to minimal risk in this trial as the intervention program compromises a lifestyle modification program through education and motivational strategies.

## Discussion

This manuscript describes the protocol for a cluster randomised control trial aiming at promoting a healthy lifestyle, and ultimately preventing the development of T2D and GDM in adult women of reproductive age in Kisantu, DRC.

Lifestyle and health promotion interventions have already been conducted in LMICs. A diabetes community lifestyle improvement programme, an RCT conducted in India, showed that after 3 years of intervention the incidence of diabetes was reduced significantly more in the intervention group when compared to the control group [43]. Similarly, an intervention study aiming to assess a lifestyle modification program on weight loss and metabolic syndrome risks in Thailand involving 60 women, showed that the intervention group had significantly greater weight loss than the control group, as well as greater improvements in blood pressure, fasting plasma glucose and waist circumference, among other indicators [44]. Moreover, an RCT conducted in Iran involving 137 pregnant women showed a lower prevalence of GDM after lifestyle information and motivational intervention of 8 weeks of duration when compared to the control group [45]. Aligned with these promising findings, we expect the implementation of our study to be successful for thereafter scaling up the multi-component programme to capture a larger population and influence policy and practice decisions regarding the introduction

of effective health promotion programmes for primary prevention of lifestyle-related conditions.

This study is the first experimental trial using health promotion strategies for the primary prevention of T2D and GDM in DRC. Methodological strengths of this research include a power calculation for the primary outcome, randomisation and a wide range of lifestyle-related information collected by quantitative instruments. Limitations of the study design comprise that the main findings, the adherence to a healthy lifestyle, may be vulnerable to response bias due to reporting socially desirable behaviours. Moreover, for secondary outcomes, the sample size might not be sufficient to detect significant changes, however, analyses concerning these outcomes will be considered exploratory. Another limitation relates to the recruitment strategy of enrolling participants attending healthcare centres for different motives, imposing difficulties to generalise the findings to other populations, nevertheless, this recruitment strategy is considered necessary to ensure good adherence to the programme.

Dissemination plans for this study include drafting the output and findings into scientific manuscripts and submitted to peer-reviewed journals for publication. Digital reports enclosing the results will be shared with health-related entities of DRC.

## Supporting information

**S1 File. SPIRIT checklist—Recommended items to address in a clinical trial protocol and related documents.**
(PDF)

**S2 File. Approved study protocol.**
(PDF)

**S3 File. Healthy lifestyle questionnaire.**
(PDF)

## Acknowledgments

We thank the City of Antwerp for the financial support provided to the study. We thank the staff of Memisa DRC and BDOM-Kisantu for the support provided to the implementation of this study, as well as the professionals at the Health District of Kisantu, and the participating women for their invaluable contribution to this research.

## Author Contributions

**Conceptualization:** Diana Sagastume, Deogratias Katsuva Sibongwere, Olivier Kidima, Diertho Mputu Kembo, José Mavuna N'keto, Jean-Claude Dimbelolo, Dorothée Bulemfu Nkakirande, Jean Clovis Kalobu Kabundi, José L. Peñalvo.

**Formal analysis:** Diana Sagastume, José L. Peñalvo.

**Funding acquisition:** Deogratias Katsuva Sibongwere, Jean Clovis Kalobu Kabundi, José L. Peñalvo.

**Investigation:** Diana Sagastume, Deogratias Katsuva Sibongwere, Jean Clovis Kalobu Kabundi, José L. Peñalvo.

**Methodology:** Diana Sagastume, José L. Peñalvo.

**Project administration:** Olivier Kidima, Diertho Mputu Kembo, Jean Clovis Kalobu Kabundi.

**Resources:** Olivier Kidima, Diertho Mputu Kembo, José Mavuna N'keto, Jean Clovis Kalobu Kabundi, José L. Peñalvo.

**Supervision:** Diana Sagastume, Deogratias Katsuva Sibongwere, Olivier Kidima, Diertho Mputu Kembo, José Mavuna N'keto, Jean Clovis Kalobu Kabundi, José L. Peñalvo.

**Writing – original draft:** Diana Sagastume.

**Writing – review & editing:** Diana Sagastume, Deogratias Katsuva Sibongwere, Olivier Kidima, Diertho Mputu Kembo, José Mavuna N'keto, Jean-Claude Dimbelolo, Dorothée Bulemfu Nkakirande, Jean Clovis Kalobu Kabundi, José L. Peñalvo.

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
