## [Decision Letter · Decision Letter 0]

19 May 2022

PONE-D-22-10378Improving lifestyle behaviour among women in Kisantu, the Democratic Republic of the Congo: A protocol of a cluster randomised controlled trialPLOS ONE

Dear Dr. Sagastume,

Thank you for submitting your manuscript to PLOS ONE. After careful consideration, we feel that it has merit but does not fully meet PLOS ONE’s publication criteria as it currently stands. Therefore, we invite you to submit a revised version of the manuscript that addresses the points raised during the review process.

We look forward to receiving your revised manuscript.

Kind regards,

Yoshihiro Fukumoto

Academic Editor

PLOS ONE

Journal Requirements:

Reviewers' comments:

Reviewer's Responses to Questions

**Comments to the Author**

1. Does the manuscript provide a valid rationale for the proposed study, with clearly identified and justified research questions?

Reviewer #1: Yes

Reviewer #2: Yes

2. Is the protocol technically sound and planned in a manner that will lead to a meaningful outcome and allow testing the stated hypotheses?

Reviewer #1: Yes

Reviewer #2: Yes

3. Is the methodology feasible and described in sufficient detail to allow the work to be replicable?

Reviewer #1: Yes

Reviewer #2: Yes

4. Have the authors described where all data underlying the findings will be made available when the study is complete?

Reviewer #1: Yes

Reviewer #2: Yes

5. Is the manuscript presented in an intelligible fashion and written in standard English?

Reviewer #1: Yes

Reviewer #2: Yes

6. Review Comments to the Author

You may also provide optional suggestions and comments to authors that they might find helpful in planning their study.

Reviewer #1: This paper was a protocol of cRCT for improving lifestyle behavior in adult women of reproductive age. This protocol is a feasible and structured study, and has a relevant point in women.

Major comment

The aim of this study is to prevent T2D and GDM, and to develop and evaluate the multi-component health promotion program. The primary outcome of this study is ambiguous, because the primary outcome is usually hard endpoint. I recommend that the primary outcome is increasing weight or BMI, and incident of T2D or GDM, and that this secondary outcome is the adherence to healthy lifestyle, and clinical indicators. I think the objectives and outcomes in you study do not match.

Reviewer #2: This is a protocol of a cluster randomised controlled trial to improving lifestyle behaviour among women in Kisantu, the Democratic Republic of the Congo. Well wittern, organized and very intersting. The protocol follows to the SPIRIT. I have only one comment to the authors.

1. What is the rationale to set the duration of intervention as 24 months for improveing lifestyle behaviour? Changing lifestyle behaviour is usually challenging as the study participants are relcutant to accept new lifestyle behaviour and these lifestyle interventions usually reduce efficacy after finishing interventions (e.g. food intervention for hypetension etc...).

7. PLOS authors have the option to publish the peer review history of their article (what does this mean?). If published, this will include your full peer review and any attached files.

Reviewer #1: No

Reviewer #2: No

---

## [Author Response · Author response to Decision Letter 0]

1 Jun 2022

RESPONSES TO EDITORS AND REVIEWERS

We would like to thank the editors and reviewers for their comments and suggestions. 

 Additional edit (requested 06/01/2022)

1. Your ethics statement should only appear in the Methods section of your manuscript. If your ethics statement is written in any section besides the Methods, please move it to the Methods section and delete it from any other section. Please ensure that your ethics statement is included in your manuscript, as the ethics statement entered into the online submission form will not be published alongside your manuscript.

Please note that Ethics statement was also appear in the Abstract section of the manuscript.

[Response] We ha chosen to separate the previous ethics statement into two subsections of the Methods section 1) Ethics statement; 2) Confidentiality and risk (Tracked manuscript: Methods section Ethics statement and Confidentiality and risks, lines 344-353, page 17; Clean version: Methods section Ethics statement and Confidentiality and risks, lines 340-348, page 17). The current ethic statement included in the manuscript has been also modified in the online submission, and equals the ethic statement in the abstract (Tracked manuscript: Abstract, lines 67-73, page 3-4; Clean version: Abstract, lines 63-69, page 3-4). 

Additional edit (requested 06/01/2022)

1. Your ethics statement should only appear in the Methods section of your manuscript. If your ethics statement is written in any section besides the Methods, please move it to the Methods section and delete it from any other section. Please ensure that your ethics statement is included in your manuscript, as the ethics statement entered into the online submission form will not be published alongside your manuscript.

[Response] The ethics statement has been moved to the Methods section (subheading Ethics statement). (Tracked manuscript: Methods section Ethics statement, lines 344-345, page 17; Clean version: Methods section Ethics statement, lines 340-342, page 17). Also, the ethics statement entered into the online submission has been modified to equal the ethics statement included in the manuscript. 

Journal Requirements:

[Response] The manuscript and corresponding files have been modified according to PLOS ONE’s requirements.

[Response] The manuscript contains the correct information regarding the funding. The funding information in the electronic portal has been updated accordingly.

[Response] Currently, datasets have not been yet generated or analysed as this manuscript describes the study protocol of ongoing research. After the study termination, data cannot be shared publicly because it entails sensitive information, however, data generated will be available upon request to the corresponding author and only after consultation with the involved ethical committees. Both The institutional Review Board of the Institute of Tropical Medicine (contact: irb@itg.be) and the Ethical Committee of the University of Kinshasa (contact: espsec_unikin@yahoo.fr), should review the request and approve any use of the data generated from this project outside the original scope of work, and identified research teams. The data derived from this study, even if pseudonymised, and deprived of person’s identifiers, contains sensitive information such as sex, age, marital status, profession, number of children, and personal lifestyle habits from which identification cannot be ruled out. This statement has been modified in the manuscript (Tracked version: section Metadata, lines 30-38, page 

2; Clean version: Section Methods/Design, lines 29-36, page 2).

[Response] For this study, no data has been generated or analysed as this manuscript only describes the study protocol of ongoing research. 

Reviewers comments

Reviewer #1: This paper was a protocol of cRCT for improving lifestyle behavior in adult women of reproductive age. This protocol is a feasible and structured study, and has a relevant point in women.

Major comment

The aim of this study is to prevent T2D and GDM, and to develop and evaluate the multi-component health promotion program. The primary outcome of this study is ambiguous, because the primary outcome is usually hard endpoint. I recommend that the primary outcome is increasing weight or BMI, and incident of T2D or GDM, and that this secondary outcome is the adherence to healthy lifestyle, and clinical indicators. I think the objectives and outcomes in you study do not match.

[Response] We thank the reviewer for this comment. The scope of this study is, indeed, to instil healthy habits in women to ultimately prevent the onset of chronic conditions, particularly type 2 diabetes and gestational diabetes whose prevalence is on the rise in DRC. The study focuses on health promotion rather than disease prevention, approaching public health from an upstream perspective and hoping to counteract the early adoption of risk factors (suboptimal diet and physical activity levels) right at the start of their adoption. As such, to evaluate the intervention’s impact, our primary outcome should be a measure of our intervention target; increase adherence to a healthy lifestyle in women. We agree that measuring the adherence to a healthy lifestyle is less objective than a hard endpoint such as diagnosis of type 2 diabetes, however, we have chosen it to effectively evaluate the impact of our intervention. As a health promotion intervention, our target population is not limited to women with overweight or obesity; women with a normal weight can also be part of the study. Using the change in weight or BMI as a primary outcome would therefore imply that our intervention carries a major component of weight maintenance/reduction which is not the case, as our intervention components revolve around instilling healthy habits. Reinforcing the potentially greater effectiveness of health promotion interventions in our research context is also the lack of information on the prevalence of metabolic risk factors (increased BMI, hyperglycemia) in this population. By choosing a health promotion approach we will introduce positive concepts, focusing on keeping health rather than developing a disease, that increase the overall wellbeing of the population while allowing us to also gain the needed insight into the frequency of metabolic risk factors. Nevertheless, we are aware of the potential introduction of response bias in our primary outcome due to socially desirable bias but this bias should be greatly minimized by the introduction of the randomized control (Tracked version: section Discussion, lines 381-383, page 18; Clean version: Section Methods/Design, line 376-378, page 18).

Reviewer #2: This is a protocol of a cluster randomised controlled trial to improving lifestyle behaviour among women in Kisantu, the Democratic Republic of the Congo. Well wittern, organized and very interesting. The protocol follows to the SPIRIT. I have only one comment to the authors.

1. What is the rationale to set the duration of intervention as 24 months for improving lifestyle behaviour? Changing lifestyle behaviour is usually challenging as the study participants are relcutant to accept new lifestyle behaviour and these lifestyle interventions usually reduce efficacy after finishing interventions (e.g. food intervention for hypetension etc...).

[Response] We thank the reviewer for the comment. We have set the duration of the programme to 24 months to guarantee we have the sufficient time to witness an improvement in lifestyle in the intervened clusters. This extended duration is uncommon in randomized trials, that tend to be of shorter duration and evaluate more intense interventions. In our case the cluster randomization allows us to maintain our control centers receiving usual care in a separate setting, offering the opportunity to implement a sustainable, context-based intervention for health promotion hoping for a gradual adoption of healthier habits. Throughout the 24 months and multiple follow up visits, we expect to see a gradual increase in healthy lifestyle measurement, in contrast with the controls that should remain constant. This gradual approach has been identified as the one with most potential to be adopted by the community and to be further uptake by health agencies in DRC, and as such as been backed up by the health district of Kisantu, DRC, and the national diabetes program (represented by Dr. Jean-Claude Dimbelolo in this work), as well as by the study sponsor and the Institute of Tropical Medicine’s IRB. (Tracked version: section Methods/Design – Quality assurance, lines 284-285, page 14; Clean version: Section Methods/Design, line 280-281, page 14). We expect this 24-month programme to be successful and thereafter for the health district of Kisantu investing in its continuation and integration into usual care. Additionally, we also aim at acquiring future funding for continuing this research, mainly related to evaluating the sustainability of the programme, scaling up the programme to target larger populations, and looking for additional avenues to maintain or increase the expected impact.

---

## [Decision Letter · Decision Letter 1]

25 Jun 2022

PONE-D-22-10378R1Improving lifestyle behaviours among women in Kisantu, the Democratic Republic of the Congo: A protocol of a cluster randomised controlled trial PLOS ONE

Dear Dr. Sagastume,

Thank you for submitting your manuscript to PLOS ONE. After careful consideration, we feel that it has merit but does not fully meet PLOS ONE’s publication criteria as it currently stands. Therefore, we invite you to submit a revised version of the manuscript that addresses the points raised during the review process.

We look forward to receiving your revised manuscript.

Kind regards,

Yoshihiro Fukumoto

Academic Editor

PLOS ONE

Journal Requirements:

Reviewers' comments:

Reviewer's Responses to Questions

**Comments to the Author**

1. Does the manuscript provide a valid rationale for the proposed study, with clearly identified and justified research questions?

Reviewer #1: Yes

Reviewer #2: Yes

2. Is the protocol technically sound and planned in a manner that will lead to a meaningful outcome and allow testing the stated hypotheses?

Reviewer #1: Yes

Reviewer #2: Yes

3. Is the methodology feasible and described in sufficient detail to allow the work to be replicable?

Reviewer #1: Yes

Reviewer #2: Yes

4. Have the authors described where all data underlying the findings will be made available when the study is complete?

Reviewer #1: Yes

Reviewer #2: Yes

5. Is the manuscript presented in an intelligible fashion and written in standard English?

Reviewer #1: Yes

Reviewer #2: Yes

6. Review Comments to the Author

You may also provide optional suggestions and comments to authors that they might find helpful in planning their study.

Reviewer #1: Authors returned to our comments, but you did not add to reasons or rationale in this manuscript.

Reviewer 2 has one comment to the authors “what is the rationale to set the duration of intervention as 24 months for improving lifestyle behavior?” I recommend that authors add the lines, “we have set the duration of the programme to 24 months to guarantee we have the sufficient time to witness an improvement in lifestyle in the intervened clusters.” in Evaluation design (p6).

I think the objectives and outcomes in your study do not match. I do not change my comment. I recommend that the primary outcome is increasing weight or BMI, and incident of T2D or GDM, and that this secondary outcome is the adherence to healthy lifestyle, and clinical indicators.

Authors described in the comments that “this study focuses on health promotion rather than disease prevention, ---- risk factors (increased BMI, hyperglycemia) in this population.” I recommend that authors state the reasons in outcome, in page 11.

Reviewer #2: The authors addessed the comments appropreately and revised manuscript improved significantly. I do not have further comments to the authors.

7. PLOS authors have the option to publish the peer review history of their article (what does this mean?). If published, this will include your full peer review and any attached files.

Reviewer #1: No

Reviewer #2: No

---

## [Author Response · Author response to Decision Letter 1]

1 Jul 2022

RESPONSES TO EDITORS AND REVIEWERS

We would like to thank the editors and reviewers for their comments and suggestions.

Journal Requirements:

[Response] We have reviewed the reference list and ensured it is complete and correct. Also we have ensured that none of the current cited references have been retracted. The following modifications were made, aside from formatting:

- Previous reference 9 was deleted as it was unrelated to the paper.

- Previous reference 18, currently 17, has been replaced with a relevant current reference.

- Previous references 25 and 28 were (previous submission) were deleted as they were duplicated references, just in another language. 

We believe the current list is correct. Please, let me know if any other modification/update of the references is needed.

Reviewers comments

Reviewer #1:

Authors returned to our comments, but you did not add to reasons or rationale in this manuscript.

Reviewer 2 has one comment to the authors “what is the rationale to set the duration of intervention as 24 months for improving lifestyle behavior?” I recommend that authors add the lines, “we have set the duration of the programme to 24 months to guarantee we have the sufficient time to witness an improvement in lifestyle in the intervened clusters.” in Evaluation design (p6).

I think the objectives and outcomes in your study do not match. I do not change my comment. I recommend that the primary outcome is increasing weight or BMI, and incident of T2D or GDM, and that this secondary outcome is the adherence to healthy lifestyle, and clinical indicators.

Authors described in the comments that “this study focuses on health promotion rather than disease prevention, ---- risk factors (increased BMI, hyperglycemia) in this population.” I recommend that authors state the reasons in outcome, in page 11.

[Response] We thank the reviewer for these comment. As suggested, we have included the rationale for setting the duration of the programme to 24 months in the revised manuscript (Tracked version: section Methods/Evaluation Design, lines 136-138, page 6; Clean version: Section Methods/Evaluation, line 138-140, page 7).

We really appreciate the thoughts of the reviewer on in the paper because it has make us realize that our objectives and outcomes were not clear enough. Therefore we have clarified the objectives (Tracked version: section Methods/Objectives, lines 118-123, page 6; Clean version: Section Methods/Objectives, line 120-125, page 6) and the reasons for choosing adherence to a healthy lifestyle as the main outcome in the manuscript (Tracked version: section Methods/Outcome, lines 211-217, page 11; Clean version: Section Methods/Outcome, line 212-218, page 12). We hope we have solved any concerns regarding our objectives and primary outcome. Also important to mention that we are targeting an apparently healthy population, therefore we do not expect them to develop any of these metabolic conditions to a great extent in the allocated follow-up. Moreover, modifying the primary outcome is not feasible anymore as the study has been already registered in ClinicalTrials.gov (NCT05039307) and has begun in October 2021. The ongoing study includes a specific sample size calculation of 288 women to detect a significant change in the stated primary outcome with an appropriate statistical power. Nevertheless, reflecting based on the reviewers comments, we have realised the need of a longer study/longer follow-up or a second trial to observe the sustainability of the intervention and capture these long(er)-term hard outcomes (e.g. incidence of T2D) proposed by the reviewer, as they are of great relevance for public health strategies. 

Reviewer #2: 

The authors addessed the comments appropreately and revised manuscript improved significantly. I do not have further comments to the authors.

[Response] We thank the reviewer for this comment.

---

## [Decision Letter · Decision Letter 2]

12 Jul 2022

PONE-D-22-10378R2Improving lifestyle behaviours among women in Kisantu, the Democratic Republic of the Congo: A protocol of a cluster randomised controlled trialPLOS ONE

Dear Dr. Sagastume,

Thank you for submitting your manuscript to PLOS ONE. After careful consideration, we feel that it has merit but does not fully meet PLOS ONE’s publication criteria as it currently stands. Therefore, we invite you to submit a revised version of the manuscript that addresses the points raised during the review process.

We look forward to receiving your revised manuscript.

Kind regards,

Yoshihiro Fukumoto

Academic Editor

PLOS ONE

Journal Requirements:

Reviewers' comments:

Reviewer's Responses to Questions

**Comments to the Author**

1. Does the manuscript provide a valid rationale for the proposed study, with clearly identified and justified research questions?

Reviewer #1: Yes

Reviewer #3: Yes

2. Is the protocol technically sound and planned in a manner that will lead to a meaningful outcome and allow testing the stated hypotheses?

Reviewer #1: Yes

Reviewer #3: Yes

3. Is the methodology feasible and described in sufficient detail to allow the work to be replicable?

Reviewer #1: Yes

Reviewer #3: Yes

4. Have the authors described where all data underlying the findings will be made available when the study is complete?

Reviewer #1: Yes

Reviewer #3: No

5. Is the manuscript presented in an intelligible fashion and written in standard English?

Reviewer #1: Yes

Reviewer #3: Yes

6. Review Comments to the Author

You may also provide optional suggestions and comments to authors that they might find helpful in planning their study.

Reviewer #1: I do not have further comments to the authors. The authors addressed back in response to our comments. This paper is better improved.

Reviewer #3: In this study protocol, a two-arm cluster randomized-controlled trial is being proposed to educate women about healthy lifestyles and to ultimately prevent the onset of type 2 diabetes and gestational diabetes. The primary outcome is adherence to a healthy lifestyle as measured by a questionnaire. Secondary outcomes will include anthropometric measures and level of physical activity.

Minor revisions:

1- Abstract: Clarify the statement: “Data will be summarized and quantity using statistical mixed models.” What statistical methods will be used to summarize the data? What outcomes will be modeled using mixed linear regression analyses?

2- Line 337: Typographical error: drop-outs

3- Line 338: Remove the word p-value from this line. Simply state “significance (one-sided), ...”

4- Line 345: If the data is not normally distributed, differences in the central tendencies rather than means is typically investigated. If the data is categorical, chi-square or Fisher’s exact tests are typically used to compare differences between groups.

5- Line 351: Indicate the type of covariance structure that will be used in the mixed model and/or the criteria for selecting it.

6- Ling 358: Drop “probability of” from this line.

7- Line 358: Consider replacing the sentence beginning with, “If multiple comparisons” with “Multiple comparisons will be adjusted by the Bonferroni method.”

8- Indicate what software is being used to capture the data.

7. PLOS authors have the option to publish the peer review history of their article (what does this mean?). If published, this will include your full peer review and any attached files.

Reviewer #1: No

Reviewer #3: No

---

## [Author Response · Author response to Decision Letter 2]

19 Jul 2022

We thank the reviewers and editorial team for their comments and suggestions. 

Reviewer #1:

I do not have further comments to the authors. The authors addressed back in response to our comments. This paper is better improved.

[Response] We thank the reviewer for the comment. 

Reviewer #3: 

In this study protocol, a two-arm cluster randomized-controlled trial is being proposed to educate women about healthy lifestyles and to ultimately prevent the onset of type 2 diabetes and gestational diabetes. The primary outcome is adherence to a healthy lifestyle as measured by a questionnaire. Secondary outcomes will include anthropometric measures and level of physical activity.

[Response] We appreciate the reviewer’s comment. 

Minor revisions:

1- Abstract: Clarify the statement: “Data will be summarized and quantity using statistical mixed models.” What statistical methods will be used to summarize the data? What outcomes will be modeled using mixed linear regression analyses?

[Response] Thanks for pointing this out. We have clarified how the data will be summarised and which outcomes will be quantified by the mixed models in the revised manuscript (Tracked version: section Abstract, lines 62-64, page 3; Clean version: Section Abstract, line 62-65, page 3). 

2- Line 337: Typographical error: drop-outs

[Response] The typographical error has been corrected in the revised manuscript (Tracked version: section Methods/Power calculation, line 334, page 16; Clean version: Section Methods/Power calculation, line 338, page 17).

3- Line 338: Remove the word ‘p-value’ from this line. Simply state “significance (one-sided), ...”

[Response] The word ‘p-value’ has been removed in the revised manuscript (Tracked version: section Methods/Power calculation, line 335, page 16; Clean version: Section Methods/Power calculation, line 339, page 17).

4- Line 345: If the data is not normally distributed, differences in the central tendencies rather than means is typically investigated. If the data is categorical, chi-square or Fisher’s exact tests are typically used to compare differences between groups.

[Response] Thank you for pointing this out. Indeed, we will use chi-square for categorical data. We have addressed the comment in the revised manuscript (Tracked version: section Methods/Data analysis, line 343, page 16; Clean version: Section Methods/Data analysis, line 347, page 17).

5- Line 351: Indicate the type of covariance structure that will be used in the mixed model and/or the criteria for selecting it.

[Response] We have clarified the type of covariance structure we will use for the mixed model in the revised manuscript (Tracked version: section Methods/Data analysis, line 351, page 16; Clean version: Section Methods/Data analysis, line 355, page 17).

6- Ling 358: Drop “probability of” from this line.

[Response] The word ‘probability’ has been removed in the revised manuscript (Tracked version: section Methods/Data analysis, line 356, page 16; Clean version: Section Methods/Data analysis, line 360, page 18).

7- Line 358: Consider replacing the sentence beginning with, “If multiple comparisons” with “Multiple comparisons will be adjusted by the Bonferroni method.”

[Response] The suggested sentence has been added to the revised manuscript (Tracked version: section Methods/Data analysis, lines 356-357, page 16; Clean version: Section Methods/Data analysis, lines 360-361, page 18).

8- Indicate what software is being used to capture the data.

[Response] The software being used to capture the data has been added to the revised manuscript (Tracked version: section Methods/Data analysis, lines 358-359, pages 16-17; Clean version: Section Methods/Data analysis, lines 361-363, page 18).

---

## [Decision Letter · Decision Letter 3]

1 Aug 2022

PONE-D-22-10378R3Improving lifestyle behaviours among women in Kisantu, the Democratic Republic of the Congo: A protocol of a cluster randomised controlled trialPLOS ONE

Dear Dr. Sagastume,

Thank you for submitting your manuscript to PLOS ONE. After careful consideration, we feel that it has merit but does not fully meet PLOS ONE’s publication criteria as it currently stands. Therefore, we invite you to submit a revised version of the manuscript that addresses the points raised during the review process.

We look forward to receiving your revised manuscript.

Kind regards,

Yoshihiro Fukumoto

Academic Editor

PLOS ONE

Journal Requirements:

Reviewers' comments:

Reviewer's Responses to Questions

**Comments to the Author**

1. Does the manuscript provide a valid rationale for the proposed study, with clearly identified and justified research questions?

Reviewer #3: Yes

2. Is the protocol technically sound and planned in a manner that will lead to a meaningful outcome and allow testing the stated hypotheses?

Reviewer #3: Yes

3. Is the methodology feasible and described in sufficient detail to allow the work to be replicable?

Reviewer #3: Yes

4. Have the authors described where all data underlying the findings will be made available when the study is complete?

Reviewer #3: Yes

5. Is the manuscript presented in an intelligible fashion and written in standard English?

Reviewer #3: Yes

6. Review Comments to the Author

You may also provide optional suggestions and comments to authors that they might find helpful in planning their study.

Reviewer #3: Minor revisions: (Line numbers refer to those in the tracked changes version of revision 3.)

1. Line 64: Improve the grammar in the following sentence to make it more understandable. The term "quantity" causes confusion. "The primary and secondary outcomes will be quantity using statistical mixed models."

2. Line 342: The following is a run-on sentence. "Demographic data will be explored for differences at baseline by comparing and testing means, if categorical data chi-square statistic will be used, of all demographic and baseline data between the groups." Consider the following revision. Group differences in baseline demographics will be compared with t-tests or chi-square tests.

3. Thoroughly proofread the entire manuscript.

7. PLOS authors have the option to publish the peer review history of their article (what does this mean?). If published, this will include your full peer review and any attached files.

Reviewer #3: No

---

## [Author Response · Author response to Decision Letter 3]

2 Aug 2022

We thank the reviewer and editorial team for their comments and suggestions. 

Reviewer #3: 

Minor revisions: (Line numbers refer to those in the tracked changes version of revision 3.)

1. Line 64: Improve the grammar in the following sentence to make it more understandable. The term "quantity" causes confusion. "The primary and secondary outcomes will be quantity using statistical mixed models."

[Response] The grammar of the sentence has been modified and clarified in the revised manuscript (Tracked version: section Abstract, lines 63-64, page 3; Clean version: Section Abstract, line 64-65, page 3).

2. Line 342: The following is a run-on sentence. "Demographic data will be explored for differences at baseline by comparing and testing means, if categorical data chi-square statistic will be used, of all demographic and baseline data between the groups." Consider the following revision. Group differences in baseline demographics will be compared with t-tests or chi-square tests.

[Response] The suggested sentence has been incorporated to the revised manuscript (Tracked version: section Abstract, lines 341-344, page 16; Clean version: Section Abstract, line 345-346, page 17).

3. Thoroughly proofread the entire manuscript.

[Response] The entire manuscript has been proofread, small grammar modifications have been made throughout the manuscript.

---

## [Decision Letter · Decision Letter 4]

30 Aug 2022

Improving lifestyle behaviours among women in Kisantu, the Democratic Republic of the Congo: A protocol of a cluster randomised controlled trial

PONE-D-22-10378R4

Dear Dr. Sagastume,

We’re pleased to inform you that your manuscript has been judged scientifically suitable for publication and will be formally accepted for publication once it meets all outstanding technical requirements.

Kind regards,

Yoshihiro Fukumoto

Academic Editor

PLOS ONE

Additional Editor Comments (optional):

Reviewers' comments:

Reviewer's Responses to Questions

**Comments to the Author**

1. Does the manuscript provide a valid rationale for the proposed study, with clearly identified and justified research questions?

Reviewer #3: Yes

2. Is the protocol technically sound and planned in a manner that will lead to a meaningful outcome and allow testing the stated hypotheses?

Reviewer #3: Yes

3. Is the methodology feasible and described in sufficient detail to allow the work to be replicable?

Reviewer #3: Yes

4. Have the authors described where all data underlying the findings will be made available when the study is complete?

Reviewer #3: Yes

5. Is the manuscript presented in an intelligible fashion and written in standard English?

Reviewer #3: Yes

6. Review Comments to the Author

You may also provide optional suggestions and comments to authors that they might find helpful in planning their study.

Reviewer #3: All comments have been addressed.

7. PLOS authors have the option to publish the peer review history of their article (what does this mean?). If published, this will include your full peer review and any attached files.

Reviewer #3: No

---

## [Editor Report · Acceptance letter]

31 Aug 2022

PONE-D-22-10378R4 

Improving lifestyle behaviours among women in Kisantu, the Democratic Republic of the Congo: a protocol of a cluster randomised controlled trial 

Dear Dr. Sagastume:

I'm pleased to inform you that your manuscript has been deemed suitable for publication in PLOS ONE. Congratulations! Your manuscript is now with our production department. 

Kind regards, 

on behalf of

Dr. Yoshihiro Fukumoto 

Academic Editor

PLOS ONE